# Patterns of Epigenetic Diversity in Two Sympatric Fish Species: Genetic vs. Environmental Determinants

**DOI:** 10.3390/genes12010107

**Published:** 2021-01-16

**Authors:** Laura Fargeot, Géraldine Loot, Jérôme G. Prunier, Olivier Rey, Charlotte Veyssière, Simon Blanchet

**Affiliations:** 1Centre National de la Recherche Scientifique (CNRS), Université Paul Sabatier (UPS), Station d’Ecologie Théorique et Expérimentale, UMR 5321, F-09200 Moulis, France; jerome.prunier@gmail.com; 2CNRS, UPS, École Nationale de Formation Agronomique (ENFA), UMR 5174 EDB (Laboratoire Évolution & Diversité Biologique), 118 route de Narbonne, F-31062 Toulouse CEDEX 4, France; geraldine.loot@univ-tlse3.fr (G.L.); veyssiere.charlotte@gmail.com (C.V.); 3Université Paul Sabatier (UPS), Institut Universitaire de France (IUF), F-75231 Paris CEDEX 05, France; 4CNRS, Interaction Hôtes-Parasites-Environnements (IHPE), UMR 5244, F-66860 Perpignan, France; olivier.rey@univ-perp.fr

**Keywords:** genetic structure, empirical comparative study, DNA methylation, nongenetic heredity, population genomics, freshwater

## Abstract

Epigenetic components are hypothesized to be sensitive to the environment, which should permit species to adapt to environmental changes. In wild populations, epigenetic variation should therefore be mainly driven by environmental variation. Here, we tested whether epigenetic variation (DNA methylation) observed in wild populations is related to their genetic background, and/or to the local environment. Focusing on two sympatric freshwater fish species (*Gobio occitaniae* and *Phoxinus phoxinus*), we tested the relationships between epigenetic differentiation, genetic differentiation (using microsatellite and single nucleotide polymorphism (SNP) markers), and environmental distances between sites. We identify positive relationships between pairwise genetic and epigenetic distances in both species. Moreover, epigenetic marks better discriminated populations than genetic markers, especially in *G. occitaniae*. In *G. occitaniae*, both pairwise epigenetic and genetic distances were significantly associated to environmental distances between sites. Nonetheless, when controlling for genetic differentiation, the link between epigenetic differentiation and environmental distances was not significant anymore, indicating a noncausal relationship. Our results suggest that fish epigenetic variation is mainly genetically determined and that the environment weakly contributed to epigenetic variation. We advocate the need to control for the genetic background of populations when inferring causal links between epigenetic variation and environmental heterogeneity in wild populations.

## 1. Introduction

Describing and understanding spatial patterns of intraspecific diversity in natural populations constitutes the basis for predicting the evolutionary dynamics of populations. So far, most studies have focused on spatial patterns of intraspecific genetic diversity [1,2], using neutral and/or nonneutral (or adaptive) molecular markers. Neutral markers are influenced by mutation, drift and gene flow, and are not directly associated to individual fitness. Nonneutral markers are influenced not only by the same processes but also by natural selection associated to the surrounding environment; they are hence associated to the fitness and adaptation of organisms [3]. Recently, there has been an increasing interest in documenting the distribution of intraspecific epigenetic diversity in wild populations because it may also have a role in adaptive potential of organisms [4]. Epigenetic variation is a major potential source of adaptive variation since it can be directly sensitive to the environment and transmitted across generations [5,6,7]. In particular, epigenetic variation may allow for the rapid adaptation of populations to changing environments, at a pace actually higher than adaptation by natural selection on standing genetic variation [5,8,9,10,11]. In this context, an important question concerns the spatial covariation that may exist between genetic and epigenetic diversity patterns in natural populations, i.e., whether genetic and epigenetic variants follow similar spatial patterns across landscapes or not. Answering this question allows testing whether these two markers carry distinct/complementary pieces of information, and speculating about their relative roles in the adaptive potential of organisms across spatial scales. This question is not trivial, as the inherent characteristics of genetic and epigenetic marks can lead to opposite predictions regarding the spatial covariation of epigenetic and genetic diversity patterns.

On the one side, some epigenetic marks are directly sensitive to external environmental cues (epimutations can be triggered by the surrounding environment at a lifetime scale), which may lead to uncorrelated genetic and epigenetic diversity patterns since genetic mutations are not directly sensitive to the environment. Indeed, environmental constraints (e.g., contaminants, diet, social stimuli, etc.) experienced by individuals along their life can alter the distribution of epigenetic marks across the whole genome [12,13,14,15]. Some of these marks induced by the environment can be transmitted across generations (inherited), and in that case, they are comparable to nonneutral genetic variance, except that (i) the mutation rate of epigenetic markers is higher and (ii) these marks are less stable over the long time [7,16,17,18]. Consequently, the epigenome (all of individual epigenetic marks on the DNA sequence) could theoretically carry a footprint of the contemporary (biotic and abiotic) environment in which the last few generations have lived. Epigenetic marks are hence expected to transmit (environmental) information that is not necessarily transmitted by (and that is hence complementary to) genetic marks [19]. In this situation, we can predict that spatial patterns of epigenetic diversity should deviate from those documented for neutral and nonneutral genetic markers. In particular, epigenetic diversity patterns should be strongly linked to environmental heterogeneity, whereas this should be less the case for nonneutral genetic diversity patterns, and obviously not the case for neutral genetic diversity.

On the other side, there is mounting evidence that alternative mechanisms can generate correlated patterns of epigenetic and (neutral) genetic diversity across natural landscapes. A first alternative hypothesis rests upon the assumption that epigenetic marks depend (either completely or partially) on genetic variation rather than on environmental variation [20,21,22], since an individual transmitting a given genetic allele during mitosis also transmits the epigenetic information carried by this allele, i.e., the epiallele [20]. Therefore, there would be a physical link between alleles and epialleles. Moreover, different types of genetically encoded molecules are required to modulate the expression of genes, such as RNA or proteins [23,24]. These molecules are involved in the establishment and stability of histone tail modifications or DNA methylation across generations [16,25]. Consequently, the establishment and stability of epigenetic marks is allowed by genetic information. A second alternative hypothesis hence states that the same neutral processes (drift, mutation, and gene flow) can influence both genetic and epigenetic markers in a similar direction [26,27,28]. Indeed, epi-mutations have been reported to occur naturally in wild populations [29] and age-related methylation drift is known to reflect imperfect maintenance of epigenetic marks through cell renewal [30]. These two hypotheses both suggest that, under certain circumstances, epigenetic and neutral genetic diversity patterns could actually strongly covary spatially across natural landscapes.

Documenting and understanding the joint distribution of genetic and epigenetic marks in natural populations is essential to tease apart the potential role of epigenetic and genetic backgrounds for the adaptive potential of populations, and a few studies have paved the way toward such an objective [26]. Up to now, these studies have led to contrasting and context-dependent results. For instance, a significant correlation was detected between genetic and epigenetic differentiation among natural population pairs of *Hordeum brevisubulatum* [31], whereas it was not the case in *Vitex negundo var. heterophylla* [32]. These contrasted patterns have also been observed in animal species [33,34], although most previous studies have focused on plant populations. At a first glance, it therefore seems that no generalities can be drawn about the spatial covariation between epigenetic and genetic diversity patterns in natural populations. We argue that an insightful way to tackle this question is to study species living in sympatry within a single landscape. Indeed, by “controlling” for the common environment, it would become possible to test whether the epigenetic response of populations to the same local environment is species-dependent or predictable across species, and to test the causal link between genetic and epigenetic marks. To date, these kinds of empirical comparative study are scarce [35,36], while they may be very helpful for generalizing findings across species.

The general objective of this study was to generate novel insights into the spatial patterns of genetic and epigenetic diversity of wild populations, and to empirically test the link between epigenetic population structure and the local environment. We conducted an empirical “comparative” study involving two sympatric freshwater fish species (*Gobio occitaniae* and *Phoxinus phoxinus*) in a common riverscape to gain insights into the links between genetic diversity, epigenetic diversity, and the environment. By focusing on the same set of sites for the two species, we tested the correlation between genetic and epigenetic diversity structure within each species by controlling for environmental covariation, and we compared this correlation between species. Furthermore, we used both supposedly neutral (microsatellites) and nonneutral (single nucleotide polymorphism, SNP) genetic markers, to gain further insights into the processes sustaining patterns of epigenetic diversity in wild populations. Because the two fish species belong to the same trophic level and display similar life history traits (similar generation time, for instance), we expected similar patterns for the two species. In particular, assuming that epigenetic marks are under partial genetic control [11,20,37], we predicted a positive and significant correlation between pairwise genetic and epigenetic differentiation for the two species, irrespectively of the type (neutral or not) of genetic marker. An absence of significant correlation would indirectly indicate that epigenetic diversity is controlled by other factors, i.e., the environment. We further tested the correlation between pairwise epigenetic (and genetic) differentiation and environmental distances between sites to quantify to which extent epigenetic diversity was determined by the local environment. For the two species, we expected that populations living in strongly distinct habitats would be highly differentiated epigenetically, which should not be observed for neutral (microsatellite) genetic differentiation, and only partially observed in nonneutral (SNP) markers. Given their different ecological requirements, we finally expected that the environmental component of epigenetic differentiation, if any, would not be driven by the same environmental factors in the two species.

## 2. Materials and Methods

### 2.1. Sampling and Site Selection

#### 2.1.1. Biological Models

The two focal fish species belong to the cyprinidae family: *Gobio occitaniae* (the Occitan gudgeon) and *Phoxinus phoxinus* (the European minnow). These two species are phylogenetically related, they belong to the same trophic level, they have a similar generation time (2–3 years), and they face similar selective pressures as they coexist in sympatry in many areas [38]. Nonetheless, they slightly differ ecologically since *G. occitaniae* is ubiquitous over a large part of the whole upstream-downstream gradient in rivers, whereas *P. phoxinus* is more specialized and lives preferentially in upstream areas. In addition, and despite the fact that they are both insectivorous, *G. occitaniae* feeds preferentially on the bottom, whereas *P. phoxinus* feeds mainly in the water column. Finally, *G. occitaniae* is larger in body length than *P. phoxinus* (mean body length at adult size is ~80–150 and ~50–90 mm, respectively).

#### 2.1.2. Study Area and Sampling

Based on a priori knowledge [39], we sampled the two fish species in 13 sites from the Garonne River basin (South-Western France, Figure 1). Sites varied according to key abiotic factors to optimize the likelihood of detecting unique epigenetic marks in these populations. In particular, we selected sites varying according to two key environmental variables directly affecting fish fitness and populations [39,40,41]: mean annual temperature (ranging from 16.4 to 23.3 °C, Table A1, see also Figure 1) and oxygen saturation (ranging from 77.5% to 114.7%, Table A1). Electric fishing was conducted during summer 2014 and performed under the authorization of “Arrêté Préfectoraux” delivered by the “Direction Départementale des Territoires” of each administrative department (Ariège, Aveyron, Haute-Garonne, Hautes-Pyrénées, Lot and Tarn et Garonne). We sampled 24 fish per species in each site, leading to a total of 312 individuals per species (*n* = 624). Fish were treated in accordance to the European Communities Council Directive (2010/63/EU) regarding the use of animals in Research, French Law for Animal Protection R214-87 to R214-137. Although DNA methylation diversity can show tissue-specific differences within an individual [42,43,44], we favored a non-lethal approach and hence a small piece of pelvic fin was sampled on each individual. It is noteworthy that, in fish, the shape and color of fins can be linked to abiotic environmental conditions [45,46]. All individuals were anaesthetized using benzocaine before fin clips. Each fin tissue was preserved in 70% ethanol for further genetic and epigenetic analyses. All individuals were released in their respective sampling site.

### 2.2. Environmental Data

All sites were characterized for 14 variables related to physicochemical characteristics (overall water quality) and river topography so as to test for association between epigenetic (and genetic) markers and environment (see Table A1). Topographical variables included river flow (m^3^·s^−1^), river width (m), river slope (%), and altitude (m) and were retrieved from the French Theoretical Hydrological Network (RHT) [47]. Physicochemical characteristics included concentrations in nitrate, nitrite, orthophosphate, and oxygen (mg·L^−1^), biological oxygen demand (BOD, mg·L^−1^), water conductivity (mS.cm^−1^), pH, suspended matter (SM, mg·L^−1^), oxygen saturation (%), and temperature (°C). They were obtained from the databases of the Water Information System of the Adour Garonne basin (SIEAG “Système d’Information sur l’Eau du Bassin Adour Garonne”; http://adour-garonne.eaufrance.fr). Here, we used values measured in July from 2013 to 2015, to take into account interannual variability and potential measurement errors. Values were averaged (for each parameter) across this period. July was chosen as the reference month since this is a period in which environmental constraints are likely to be strong on fish fauna (low water level, hyperthermia, hypoxia, etc.) and because it is the most informed month in the SIEAG database. All these variables are known to affect dynamics of fish populations and properly characterize the environmental conditions encountered by fish [38,48].

A principal component analysis (PCA) was performed on the 14 environmental variables using the R package “ade4” [49], to synthetize data into orthogonal variables. The three first axes represented 71.96% of the total variance (Table 1 for details), and were hence retained as synthetic environmental variables. The first axis, defined by a strong contribution of (in decreasing order) oxygen concentration, water conductivity, nitrite concentration, oxygen saturation, and nitrate concentration (Table 1), stands for a eutrophication gradient. Sites with positive values along this axis were characterized by a low concentration of oxygen, a high conductivity and high concentrations in nitrate and nitrite (i.e., the more eutrophic sites). The second axis, defined by a strong contribution of river flow, river width, and pH (Table 1), stands for an upstream–downstream gradient. Sites with positive values along this axis were characterized by a large river bed (high water flow) and high pH values. The third axis is defined by a strong contribution of orthophosphate concentration, slope, altitude, and suspended matter (Table 1). Sites with positive values along this axis were characterized by high altitude sites with a steep slope and high values of nutrient and suspended matter.

### 2.3. Genetic and Epigenetic Data

#### 2.3.1. Genetic Data

The DNA of all individuals (*n* = 624) was extracted using a salt-extraction protocol [50]. Individual genetic data consisted in both microsatellite (supposedly neutral) and SNP markers (potentially nonneutral).

Microsatellites data (13 and 17 loci for *G. occitaniae* and *P. phoxinus*, respectively) were obtained from a previous study [39]. Details on accession numbers, conditions for polymerase chain reactions (PCRs), and preliminary analyses (e.g., search for null alleles or possible linkage disequilibrium between loci) are provided in Fourtune et al. [39].

SNP markers (1892 and 1244 in *G. occitaniae* and *P. phoxinus*, respectively) were obtained from the restriction site-associated DNA (RAD) sequencing of pooled DNAs at the site and species levels [51], using laboratory and bioinformatic procedures described in Prunier et al. [52]. As the DNA of individuals was pooled at the site level, we were unable to retrieve individual genotypes (contrary to microsatellite markers) and we therefore used the frequencies of alleles (from each SNP) at the population level as raw genomic data for the SNPs.

#### 2.3.2. Epigenetic Data

Individuals were then genotyped using Methylation-Sensitive-AFLP (MS-AFLP). MS-AFLP allows identifying “genome-wide” methylation patterns. This is a modified version of standard AFLP (Amplified Fragment Length Polymorphism) technique [53] that is well suited for nonmodel species (without a reference genome) and useful to assess epigenetic diversity for large sample sizes (>200) [37]. MS-AFLP relies on two separate double digestions with EcoRI (rare cutter, on 5’G|AATTC restriction site) and either one of two methylation-sensitive restriction enzymes (HpaII and MspI, frequent cutters, on 5’CC|GG restriction site). Because HpaII and MspI have different cytosine methylation sensitivities, comparison of the two digestion fragment profiles (EcoRI/MspI and EcoR1/HpaII) leads to the distinction of four methylation conditions for each DNA fragment: Condition I = fragments are present in both profiles, indicating an unmethylated state; Condition II = fragments are present only in EcoRI/MspI profile indicating an hemimethylation of internal cytosine (^HMe^CG-sites) or a full methylation of (both) internal cytosines (^Me^CG-sites); Condition III = fragments are present only in EcoRI/HpaII profile indicating an hemimethylation of external cytosine (^HMe^CCG-sites); Condition IV = fragments are absent in both profiles, indicating an uninformative state [54]. This last case can have multiple origins such as full methylation on external cytosine (^Me^CCG-), hemimethylation of both cytosines (^HMe^C^HMe^CG-sites), full-methylation of both cytosines (^Me^C^Me^CG-sites), or more rarely genetic mutation leading to polymorphism of the restriction site.

#### 2.3.3. MS-AFLP Protocol

The first step consists in two separate double digest reactions of 3 µL of extraction product (30–40 ng·µL^−1^) with 0.5 µL of FastDigest EcoRI (1 FDU·µL^−1^) and 0.5 µL of either FastDigest MspI (1 FDU·µL^−1^) or FastDigest HpaII (1 FDU·µL^−1^) isoschizomers. DNA was digested at 37 °C for 15 min. Double-stranded adaptors (see Table A2 for details) [32,55,56] were then ligated onto the sticky end of all the digestion products (10 µL) with 0.3 µL of a T4 DNA ligase (5 U/µL, Thermo Scientific) and 1 µL of each adaptor (EcoRI adaptors 2.5 µM; MspI/HpaII adaptor 0.25 µM) at 25 °C for 1 h. After a step of enzyme killing, the product was subjected to two rounds of increasingly selective PCR amplification (PCR1 and PCR2). Preselective amplification (PCR1, see Appendix C for details) was performed in a total volume of 25 µL using 5 µL of 5x buffer, 1.5 µL of dNTP (10 mM), 2 µL of each preselective primer (10 μM, see Table A2 for sequences), 0.3 µL of Taq DNA polymerase (5 U/µL Thermo Scientific^®^), and 2 µL of ligation product. Preamplified products were then diluted to 1:50 in sterile water. Selective amplification (PCR2, see Appendix C for details) was then performed under the same conditions (reagents and total volume) than the preselective amplification, except that three specific selective primers couples were used (see Table A2 for sequences). Primers for selective PCRs were chosen among a set of 24 and 23 primers for *G. occitaniae* and *P. phoxinus*, respectively, that we previously tested for optimal conditions (number of loci amplified per primer, not shown). Amplified products were then diluted to 1:15 in sterile water and 2.2 µL of this mix was added in 7.8 µL of a mix composed of 800 µL of Hi-Di formamide (Applied Biosystems^®^) and 15 µL of ROX500 (Applied Biosystems^®^) prior to analyzing and sizing the fragments. Fragment analysis was performed on an ABI PRISM 3730 capillary sequencer (Applied Biosystems^®^, Foster City, CA, USA) at the Génopôle Toulouse Midi-Pyrénées.

Fragment profiles were analyzed with GENEMAPPER 5.0^®^ and we scored fragments (loci) between 150 and 500 bp to avoid homoplasy [57]. Binning of fragments was performed using a peak height threshold at 750 Relative Fluorescence Units (RFU) to exclude all ambiguous peaks. Manual verification permitted to eliminate false positive such as peaks just above or below the threshold set, fluorescence blobs, or peaks too close one from the other to be correctly resolved by automated analysis. Absence and presence of data at each locus were then converted into Conditions I, II, II, or IV as explained above [54]. All loci that contained Condition IV (i.e., uninformative state) for more than 95% of the individuals were excluded from further analyses. This resulted in a total of 251 polymorphic loci for *G. occitaniae* and 274 polymorphic loci for *P. phoxinus*, respectively (see the number of loci per primer in Table A2). We considered each of the four conditions as carrying unique information, and we therefore ran statistical analyses directly on a four-state data matrix, which permitted us to keep all the information contained in the dataset.

### 2.4. Statistical Analyses

To test the part of the molecular variance that was explained by the between-population component, an analysis of molecular variance (AMOVA; Excoffier et al. 1992) was performed on either genetic or epigenetic markers and for each species separately (“poppr.amova” function from the poppr R package). Regarding genetic markers, only microsatellites markers were considered here, because we did not have the within-population component (individual genotypes) in our SNPs dataset (see above). If epigenetic marks are more sensitive to the environment, they should be more discriminant and the between-population component should be higher for epigenetic markers than for genetic markers.

We then estimated measures of genetic (for both marker types separately) and epigenetic differentiation (for each species separately) by calculating the Gst’’ index of differentiation between each pair of populations. We preferred this metric of differentiation over other metrics (e.g., Fst, Gst, Jost’s D, etc.) as it has been shown to be robust to variations in mutation rates and sample sizes [58,59].

To test whether pairwise epigenetic differentiation was dependent upon genetic differentiation (i.e., whether epigenetic differentiation was genetically determined), a simple Mantel test was first performed between pairwise genetic and epigenetic distances for each species separately and for each genetic marker type separately (“mantel.rtest” function from the ade4 R package). Simple Mantel tests were also used to assess the significance of the correlation between pairwise differentiation measured from microsatellite markers and differentiation measured from SNP markers.

To test whether epigenetic differentiation between populations resulted from environmental differences among sites (i.e., whether epigenetic differentiation was environmentally determined), simple Mantel tests were also performed between either genetic or epigenetic pairwise distances and each of the three environmental distance matrices computed from retained principal components (Euclidian distances) and a geographical distance matrix based on riparian distance between sites (to control for a potential confounding spatial effect and to test for patterns of isolation-by-distance). To further investigate the relationship between epigenetic pairwise differentiation and environmental or geographical distance matrices, multiple regressions on distance matrices (MRM, “MRM” function from the ecodist R package) were then performed. MRM is an extension of partial Mantel test allowing to test the relationship between a response matrix and any number of explanatory matrices, where each matrix contains distance or similarities (Smouse et al. 1986). For each species, the pairwise matrix of epigenetic differentiation was the response variable, and explanatory variables where the three environmental distance matrices, the geographical distance matrix, and the pairwise matrix of genetic differentiation based on SNP markers to account for a possible genetic determinism of epigenetic marks. For the sake of simplicity, we did not include the pairwise matrix of genetic differentiation based on microsatellites, although results were very similar whether we integrated it or not in the models (not shown).

## 3. Results

### 3.1. Genetic and Epigenetic Discrimination

Molecular analysis of variance (AMOVA) revealed that a significant part of the genetic (microsatellite markers) and epigenetic variance was attributed to the between-population component, for both species (*p*-value < 0.001; permutation tests with 1000 repetitions). For *G. occitaniae*, the part of the total variance explained by the between-population component was twice as high for epigenetic markers as it was for genetic markers (20.15% and 10.34%, respectively, see Table 2). For *P. phoxinus*, a similar trend was observed although less pronounced (19.59% and 16.75%, respectively, see Table 2). This suggests that, in both species, epigenetic markers were more powerful to discriminate among populations than genetic markers.

### 3.2. Simple Associations between Epigenetic, Genetic, and Environmental Distances

Simple Mantel tests demonstrated that there was a significant correlation between pairwise genetic and epigenetic distance matrices in *G. occitaniae* for both microsatellite and SNP markers (*r* = 0.363, *p*-value < 0.05 and *r* = 0.531, *p*-value < 0.001, for microsatellites and SNPs, respectively; Figure 2a and Table 3). In *P. phoxinus*, although the same tendency was observed, the correlation was not significant (*r* = 0.287, *p*-value = 0.089 and *r* = 0.294, *p*-value = 0.121 for microsatellites and SNPs, respectively; Figure 2b and Table 3). Moreover, most Gst’’ values measured using epigenetic markers were above the 1:1 line, indicating that the mean pairwise differentiation among populations was higher when using epigenetic markers than when using genetic markers (Figure 2a,b). As expected, relationships between pairwise genetic distances measured using microsatellites in the one hand and SNPs on the other hand were strong and highly significant (Table 3).

In *G. occitaniae*, there was a significant relationship between epigenetic pairwise distances and environmental distances computed from the first principal component (Figure 3a and Table 4). A similar relationship was observed between pairwise genetic differentiation measured using SNPs and distance between sites measured from the same PCA axis, whereas such a relationship was not significant when considering microsatellite markers (Figure 3b and Table 4). In *G. occitaniae*, environmental distances measured from other PCA axes were not correlated to epigenetic or genetic pairwise matrices of differentiation (Table 4). In *P. phoxinus*, none of the relationships between epigenetic or genetic pairwise differentiation and environmental and geographic pairwise distances were significant (Table 4).

### 3.3. Multiple Associations between Epigenetic, Genetic, and Environmental Distances

Multiple regressions on distance matrices (MRM) revealed that, in both species, there was no relationship between epigenetic pairwise differentiation and environmental and geographic pairwise distances (Table 5). MRM showed that there was a significant relationship between genetic and epigenetic differentiation (Table 5) in *G. occitaniae*, but not in *P. phoxinus*.

## 4. Discussion

Although patterns of genetic and epigenetic structure in natural populations have already been investigated in plants [26], the relative role of genetic and epigenetic variation in driving the adaptive potential of animal populations remains unclear. Here, we compared the epigenetic structure of two sympatric freshwater fish species along the same environmental gradient, and tested the relationship between genetic diversity, epigenetic diversity, and the environment. Our results suggest that epigenetic diversity is mostly influenced by the genetic background of organisms and weakly influenced by environmental variation.

We found a tendency toward a positive correlation between pairwise genetic and epigenetic distance matrices in both species, although the correlation was significant only in one of the two species (*G. occitaniae*). This suggests that epigenetic diversity might be partly genetically controlled and/or that similar processes operate in the same manner on these two molecular markers (genetic and epigenetic). Indeed, consistent with our hypothesis, there was spatial congruency between pairwise genetic and epigenetic differentiation in *G. occitaniae*, irrespective of the type (microsatellites or SNP) of genetic marker used to assess genetic differentiation. A similar tendency, yet not significant, was observed in *P. phoxinus*, indicating that the strength of the positive association between genetic and epigenetic differentiation might slightly differ among species from the same ecological guild. A very few comparative studies in wild populations have been performed so far, and it is hence extremely difficult to draw general conclusions. In two plants species (*Spartina alterniflora* and *Borrichia frutescens*) sharing the same habitats, a positive correlation between pairwise genetic and epigenetic differentiation have been highlighted in one species (*S. alterniflora*) and not in the other (*B. frutescens*) [35]. This latter study and our findings suggest that patterns of covariation between genetic and epigenetic diversity in wild populations is likely to be species-dependent and hard to predict.

In *G. occitaniae*, the association between genetic and epigenetic differentiation was particularly strong when genetic differentiation was calculated using SNP markers. SNP markers (or some of them) are supposedly nonneutral and hence are significantly affected by natural selection, whereas microsatellites are supposedly neutral and hence mainly affected by drift and dispersal. Beyond these characteristics, epigenetic markers considered here, MS-AFLP, are likely to be more similar to SNP than to microsatellite markers in terms of the amount of evolutionary information they reveal about wild populations of fish species. Indeed, microsatellite markers are known to have a faster mutation rate and thus a higher level of polymorphism [60] than MS-AFLP and SNP markers. Consequently, the impact of neutral processes such as mutation, genetic drift, or gene flow are probably more similar between MS-AFLP and SNP markers than between MS-AFLP and microsatellite markers. In order to test the association between genetic and epigenetic differentiation, comparison between different genetic markers have previously been done in a few studies [26,31,61,62,63,64,65]. These studies also found different patterns according to the type of genetic markers that was used to estimate genetic differentiation, which confirms that the different regions of the genome are not affected by neutral processes in the same manner [66,67,68] and strongly suggests that estimating genetic differentiation based on a single marker type can potentially lead to imprecise conclusions, especially, if this latter differs in mutation rate and polymorphism level compared to epigenetic marks.

We found a significant link between epigenetic variation and environmental heterogeneity in *G. occitaniae*, but not in *P. phoxinus*. Nonetheless, when we controlled for the underlying genetic structure of populations, this link was no longer significant, suggesting a non-causal (spurious) association [69]. This indicates that, in *G. occitaniae,* the association between epigenetic diversity and environment actually occurred because of an actual causal relationship between genetic diversity and environment and a covariation at the genome level between epigenetic and genetic marks. On the contrary in *P. phoxinus*, none of these associations between environment, genetic diversity, and epigenetic diversity was uncovered. The strong and significant association between the environment and genetic diversity observed in *G. occitaniae* and the absence of such association in *P. phoxinus* is in agreement with previous findings that phenotypic differentiation between sites is strongly associated with the environment in *G. occitaniae* but not in *P. phoxinus* [39]. In particular, oxygen saturation was strongly associated with phenotypic divergence among *G. occitaniae*, and complementary analyses (Commonality Analysis, not shown) also revealed that oxygen concentration was the most impacting environmental variable associated to genetic (SNPs) differentiation. Overall, these findings are consistent with the hypothesis of natural selection, triggered by environmental stress—here oxygen—and modulated by genetic marks, resulting in a local phenotypic adaptation of *G. occitaniae* [11,39,70]. To sum up, the epigenetic-environment association found in *G. occitaniae* was actually spurious, which stresses the importance of controlling for the genetic background of populations to infer the causal link between epigenetic variations and environmental heterogeneity in wild populations.

In both species, our results highlighted that epigenetic marks were more powerful to discriminate populations than the two genetic markers. Indeed, in both species, the AMOVA revealed that the variance measured among populations was higher when using epigenetic markers than when using microsatellite markers. Moreover, using the same metric of differentiation (Gst’’), we found that the mean pairwise differentiation among populations was higher when using epigenetic markers than when using genetic markers (either microsatellites or SNPs, see Figure 2; most Gst’’ values measured using epigenetic marks are above the 1:1 line). In other words, environmentally -and geographically- distant populations were more different epigenetically than genetically. This result suggests that although a part of the epigenetic marks seems to be genetically controlled, epigenetic diversity also contained information that seems independent from genetic variation and that allows discriminating populations further. This strong discriminative power of epigenetic marks is unlikely to be mainly driven by the sensitivity of epigenetic marks to the environment as we found little evidence that, in these species, the epigenetic structure of populations was causally linked to the environment (see above). Rather, we can speculate that the inherent characteristics of epigenetic marks (in particular, higher mutation rates) and their sensitivity to neutral processes (drift and dispersal) make them extremely relevant as natural markers for population discrimination. This strong discriminative ability might be highly relevant for species conservation, for instance, to identify ecologically and evolutionary isolated populations (and hence Evolutionary Significant Units) [71,72,73] and/or infer connectivity among populations [4].

Finally, we want to address some methodological limitations to our work. First, we worked on fin tissue to favor a non-lethal approach, supported by the fact that the shape of the fin and its coloration can be linked to abiotic environmental conditions [45,46]. In this context, the fin appears to be a good compromise between both scientific and ethical concerns. However, given that DNA methylation diversity can show tissue-specific differences within an individual [42,43,44], this choice is not trivial. The fin is likely not the tissue that responds the most, at the molecular level, to environmental conditions, and is consequently probably less linked to fitness than other tissues. Our results might have been different with other tissues like muscle [33,74,75,76], blood [34,77,78], liver [79,80], or gill tissue [81]. Some authors made different choices to avoid tissue-specific differences such as gonads [56] or even whole organism when it is relatively small in body size [36,82,83,84]. Second, we used a MS-AFLP protocol, which is currently the most widely used approach for inferring epigenetic diversity in wild populations [37,78,81,85]. Although MS-AFLP has several advantages, such as being efficient and economical to assess epigenetic diversity for large sample sizes, this method only provides anonymous and dominant markers leading to a fragment analysis that is subject to homoplasy (i.e., two fragments of the same size but with different sequence) [57]. Consequently, DNA methylation marks are difficult to link to the functional context or to compare directly with genetic data [78,86]. Promising approaches based on reduced representation bisulfite sequencing (RRBS) approach and next-generation sequencing (NGS) might partly solve these issues [87,88], particularly by allowing the identification of the specific loci potentially implied in the response to the environment. RRBS should be explicitly compared to MS-AFLP to isolate further the potential limits of MS-AFLP approaches and to gain novel insights into the loci underlying adaptation to the local environment [89].

## 5. Conclusions

To conclude, our study provided an attempt to link epigenetic variation in wild populations to the surrounding environment, a work that has been almost always carried out in plants and much more rarely in animals [20,90]. In our empirical comparative study, we showed that, contrary to expectations, there was no link between epigenetic variation and environmental constraints in *G. occitaniae* and *P. phoxinus.* This suggests that epigenetic diversity might be poorly associated to adaptation in these two species. Nonetheless, in both cases, epigenetic variation seems to be genetically determined, indicating a genetic control of epigenetic variation, as suspected in previous works [20,21,22]. Interestingly, epigenetic differentiation was linked (or show a tendency to be linked) to microsatellites (i.e., neutral) genetic differentiation, reinforcing the idea of an impact of the same neutral processes on genetic and epigenetic variation [26,27,28]. This implies that, in the species we investigated, epigenetic variation is more likely driven by neutral than nonneutral processes. Nonetheless, epigenetic marks are still more efficient than genetic markers to discriminate populations and can hence provide a tool to improve conservation strategies of endangered populations [4]. Future works should hence consider the dual use of genetic and epigenetic marks to inform conservation strategies, such as the delimitation of significant units of conservation or the quantification of biological connectivity in fragmented landscapes.

## Figures and Tables

**Figure 1 genes-12-00107-f001:**
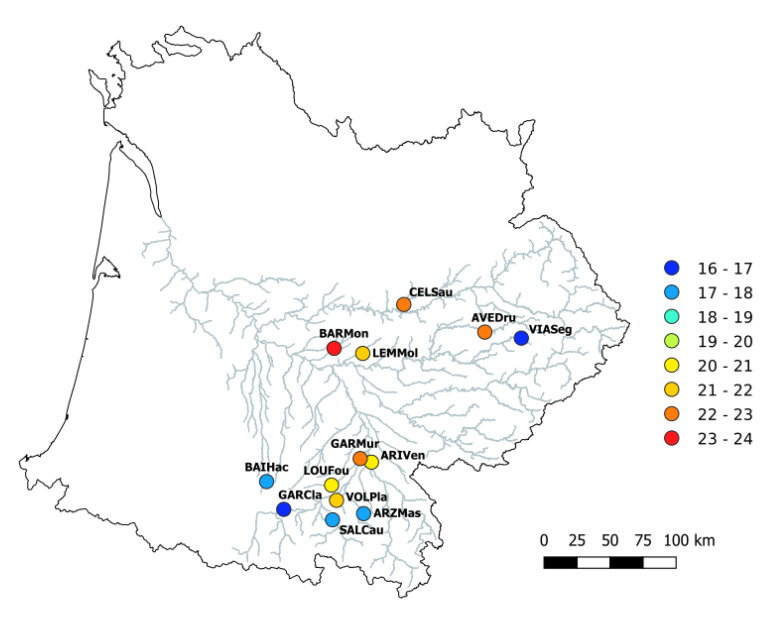
Distribution of the 13 sampling sites in the Garonne river basin. Names and localization are highlighted in bold. Twenty-four individuals per site and per species (*Gobio occitaniae* and *Phoxinus phoxinus*) have been sampled. Color of circles indicates mean water temperature (°C).

**Figure 2 genes-12-00107-f002:**
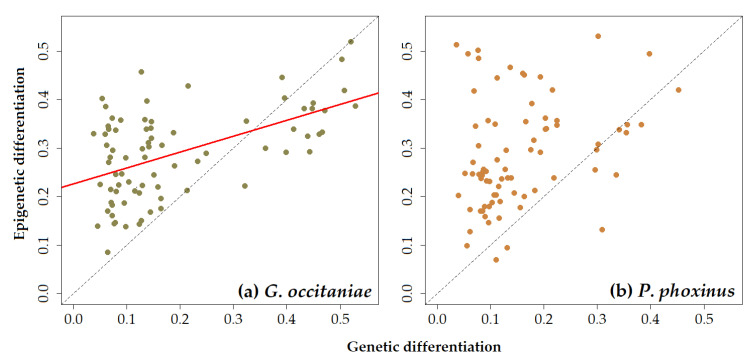
Biplots illustrating the relationships between pairwise genetic (based on SNP markers) and epigenetic differentiation (based on MS-AFLP markers) in (**a**) *G. occitaniae* (the red line indicates a significant relationship based on a simple Mantel test) and (**b**) *P. phoxinus* (no significant relationship was detected based on a simple Mantel test). Each dot represents a pairwise distance between two sites. The dashed line indicates the 1:1 line. Similar trends were observed with microsatellite markers but are not shown here.

**Figure 3 genes-12-00107-f003:**
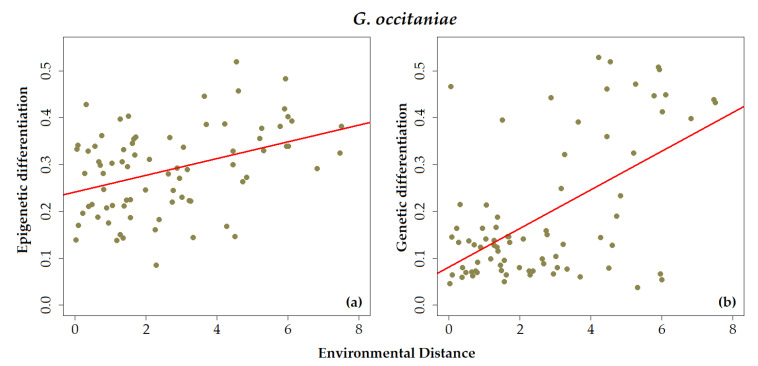
Biplot illustrating the relationship between (**a**) epigenetic differentiation between pairs of populations (based on MS-AFLP) and (**b**) genetic differentiation between pairs of populations (based on SNPs markers) and environmental distances (along a eutrophication gradient) between pairs of sites for *G. occitaniae* (the red line indicates a significant relationship based on simple Mantel test). Each dot represents a pairwise distance between two sites.

**Table 1 genes-12-00107-t001:** Characteristics of the three first principal components from the principal component analysis (PCA) ran on the 14 environmental variables and used to characterize each of the 13 sampling sites. The part of the total environmental variance (%) and the contribution of each variable to each component are shown. The variables that contributed significantly to the axis are shown in bold. BOD = biological oxygen demand; SM = suspended matter.

–	Component 1	Component 2	Component 3
**Part of total** **variance (%)**	37.03	20.87	14.06
**River flow**	−0.273	**0.888**	−0.056
**River width**	−0.295	**0.864**	0.037
**Slope**	−0.434	−0.407	**0.614**
**Altitude**	−0.538	−0.442	**0.596**
**Conductivity**	**0.878**	0.258	0.212
**BOD**	0.555	−0.020	0.045
**MS**	0.262	0.060	**0.529**
**Nitrate**	**0.748**	0.166	0.344
**Nitrite**	**0.855**	0.199	0.266
**Orthophosphate**	0.354	0.303	**0.757**
**Oxygen**	**−0.903**	0.267	0.191
**pH**	−0.341	**0.719**	0.075
**Oxygen saturation**	−**0.848**	0.304	0.238
**Temperature**	0.577	0.339	−0.208
**Global characteristic**	Oligotrophic water–Eutrophic water	Small river–Large river	Low altitude and nutrient–High altitude and nutrient

**Table 2 genes-12-00107-t002:** Outputs of analyses of molecular variance (AMOVA) aiming at testing the part of the molecular variance that was explained by the between-population component (the within-population component is not shown here). Results are presented for the two fish species (*G. occitaniae* and *P. phoxinus*) and the two molecular marker types (genetic and epigenetic markers) separately. For the genetic marker, only microsatellite markers have been considered in this analysis (see the text for details). The percentages of the total variance (“Variation”) explained by the between-population component (and the associated Phi-st values) are presented, as well as the respective *p*-values based on permutation tests with 1000 repetitions.

	Degrees of Freedom	Sum of Squares	Variance Components	Variation (%)	*Phi-st*	*p*-value
***G. occitaniae***						
Genetic markers	12	217.182	0.481	10.34	0.103	<0.001
Epigenetic markers	12	5726.943	17.210	20.15	0.202	<0.001
***P. phoxinus***						
Genetic markers	12	484.297	1.160	16.75	0.168	<0.001
Epigenetic markers	12	6369.97	19.478	19.59	0.196	<0.001

**Table 3 genes-12-00107-t003:** Summary of simple Mantel tests testing the relationships between genetic and epigenetic differentiation between pairs of populations in *G. occitaniae* and *P. Phoxinus*. Results are presented for both genetic (microsatellites and SNPs) and epigenetic (MS-AFLP) markers. Mantel statistics are presented (above diagonal), as well as the associated *p*-values based on 1000 permutations (below diagonal). Significant relationships are bolded.

	Microsatellites	SNP	MS-AFLP
***G. occitaniae***			
Microsatellites	–	**0.616**	**0.363**
SNP	**0.009**	–	**0.531**
MS-AFLP	**0.011**	**<0.001**	–
***P. phoxinus***			
Microsatellites	–	**0.894**	0.287
SNP	**<0.001**	–	0.294
MS-AFLP	0.089	0.121	–

**Table 4 genes-12-00107-t004:** Summary of simple Mantel tests testing the relationships between epigenetic and genetic pairwise differentiation, environmental (PCA components 1 to 3), and geographical (riparian) pairwise distances in *G. occitaniae* and *P. phoxinus*. Results are presented for epigenetic (MS-AFLP) and two genetic (microsatellite and SNPs) markers. Mantel statistics (r) are presented, as well as the associated *p*-values based on permutation tests with 1000 repetitions. Significant relationships are bolded.

	Component 1		Component 2		Component 3		Riparian Distance	
	r	*p*-value	r	*p*-value	r	*p*-value	r	*p*-value
***G. occitaniae***								
MS-AFLP	**0.395**	**<0.01**	0.033	0.397	−0.027	0.570	−0.060	0.670
SNP	**0.562**	**<0.05**	−0.142	0.670	−0.104	0.661	−0.117	0.666
Microsatellites	0.161	0.195	−0.066	0.538	−0.162	0.765	0.095	0.333
***P. phoxinus***								
MS-AFLP	−0.154	0.844	−0.114	0.669	−0.143	0.797	−0.080	0.639
SNP	−0.217	0.872	−0.072	0.539	−0.035	0.479	0.107	0.328
Microsatellites	−0.11	0.663	−0.180	0.747	−0.001	0.436	0.186	0.228

**Table 5 genes-12-00107-t005:** Summary of multiple regression on distance matrices (MRM) testing the relationships between epigenetic differentiation, genetic differentiation (based on SNP markers), environmental (PCA components 1 to 3), and geographical (riparian) distances between sites in *G. occitaniae* and *P. phoxinus*. Parameters associated to each explanatory variable are shown, together with their *p*–values (1000 permutations). Significant relationships are bolded.

	*G. occitaniae*		*P. phoxinus*	
	Coefficients	*p*-value	Coefficients	*p*-value
**Intercept**	0.197	0.945	0.312	0.362
**Component 1**	0.007	0.317	−0.008	0.370
**Component 2**	0.006	0.473	−0.006	0.672
**Component 3**	<0.001	0.925	−0.009	0.559
**Riparian Dist.**	<0.001	0.817	−0.001	0.534
**Genetics (SNP)**	**0.290**	**0.003**	0.307	0.220

## Data Availability

The data presented in this study are openly available in (FigShare) at (https://doi.org/10.6084/m9.figshare.13580534.v1), reference number (13580534).

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
