# Peer review of "Patterns of Epigenetic Diversity in Two Sympatric Fish Species: Genetic vs. Environmental Determinants"

_genes, 2021, doi:10.3390/genes12010107_

Round 1

Reviewer 1 Report

This manuscript is generally well written and I have only a few suggestions regarding style (below).  The only major comment is in regards to the Discussion. I applaud the valuable points made in the Discussion on lines 375-377 (i.e. epigenetic diversity is mostly influenced by genetic background); 424-427 (i.e. importance of controlling the genetic background); and 441-446 (epigenetic marks are highly relevant for population discrimination). I also appreciated the brief discussion about the limitations of the techniques employed. That said, it would be very helpful to folks that are perhaps less acquainted with the techniques used, to provide a brief discussion of the potential differences between the neutral and non-neutral genetic markers used and how each was informative (or not informative) in slightly different ways. Each of these markers could have a different bearing on your outcomes which need to be emphasized. 

Specific edits:

Line 356: remove "aiming at".

Line 366: remove "aiming at".

Line 381:  ...same manner on these two molecular markers (microsatellites and SNPs).

Line 383: ...irrespective of...

Line 393: Remove this dangling modifier. The sentence does not need it.

Line 421: What type of complementary analyses are you specifically referring to?

Line 447: "Eventually" is an incorrect modifier for this sentence. Please remove or use "Finally" or similar.

In all, a nice paper to read.

Reviewer 2 Report

The authors present a fascinating study comparing spatial patterns of genetic and epigenetic (DNA methylation) variation for two sympatric species of fish. While epigenetic markers (methylation sensitive AFLP) better differentiated populations and correlated with axes environmental variation for one species this information was not independent of the axes of genetic variation. Indeed when controlling for spatial patterns of genetic variation, the epigenetic signatures failed to be explained by any additional environmental variance.

This paper was wonderfully conceived and thoughtfully constructed. I thoroughly enjoyed reading it. You have convinced me that the manuscript addresses a fundamentally interesting question and provides one of the first comparative analyses of its kind in fish (and vertebrates for that matter). While the MS-AFLP analyses is a relatively coarse measure of epigenetic variation, I think it a robust approach and still provides a suitable approximation of average genome-wide epigenetic variation.

I have one over-arching comment regarding the analyses. By using the average differentiation for each of the datasets (microsat, SNP, and MS-AFLP) there is an implicit assumption that environmental variables are broadly correlated with most markers across the entire genome. This may, or may not, be a reasonable assumption. As you have nicely detailed in the manuscript, loci under selection will also be influenced by the effects of mutation, drift, and gene flow. The interplay of all these forces will ultimately determine whether individual loci demonstrate associations with environment or not. I suspect that effects of selection by the studied environmental variables would need to be fairly strong to over-ride the effects of drift in determining genome-wide patterns of differentiation. To this point, it may have been more useful to identify and separate genetic loci (SNPs) with strong environmental associations independent of any demographic effects (i.e. drift and gene flow) to create a set of “environmentally” associated markers that would allow you to further tease apart whether local demographic effects or selection due to environment or plastic changes best explain epigenetic variation.

In the same vein, epigenetic loci are unlikely to homogeneously be influenced by the same forces. I don’t know the intricacies of AFLP analysis and what is possible, but again separating loci by those with or without environmental associations and then assessing to what degree certain factors influence variation of these two sets would be interesting. At the same time, it would allow you to estimate the fraction of the genome / epigenome that is influenced by genetic vs. environmental factors. While I think it would be cool to explore the data in these ways, I don’t think it is necessary to add these additional analyses as this study is already well conceived and presented.

Line 176: Do you expect a level of temporal discordance of environmental associations with epigenetic vs. genetic variation. I suspect that patterns of “adaptive” genetic variation reflect selection over a much longer period of time than for epigenetic variation. If an older time series of environmental data (last couple decades maybe) were available, do the observed patterns hold if using older vs. more recent climate / environmental data? It’s not necessary to add, but just an idea for future consideration.

Line 314: This is a good conclusion, but I think it should be made clear again here that this in comparison to only the microsatellite dataset which almost certainly lacks power relative to the SNP or MS-AFLP datasets due to the smaller number of markers.
